# Environmental Enrichment in Cancer as a Possible Tool to Combat Tumor Development: A Systematic Review

**DOI:** 10.3390/ijms242216516

**Published:** 2023-11-20

**Authors:** Matheus Santos de Sousa Fernandes, Tiago Ramos Lacerda, Débora Eduarda da Silva Fidélis, Gabriela Carvalho Jurema Santos, Tayrine Ordonio Filgueira, Raphael Fabrício de Souza, Claúdia Jacques Lagranha, Fábio S. Lira, Angela Castoldi, Fabrício Oliveira Souto

**Affiliations:** 1Instituto Keizo Asami, Universidade Federal de Pernambuco, Recife 50740-600, Pernambuco, Brazil; matheus.sfernandes@ufpe.br (M.S.d.S.F.); tayrine.ordonio@ufpe.br (T.O.F.); angela.castoldi@gmail.com (A.C.); 2Programa de Pós-Graduação em Biologia Aplicada à Saúde, Centro de Biociências, Universidade Federal de Pernambuco, Recife 50740-600, Pernambuco, Brazil; tiago.lacerdar@hotmail.com (T.R.L.); deeborafidelis_@hotmail.com (D.E.d.S.F.); 3Programa de Pós-Graduação em Nutrição, Universidade Federal de Pernambuco, Recife 50740-600, Pernambuco, Brazil; gaby9carvalho@gmail.com; 4Department of Physical Education, Federal University of Sergipe, São Cristovão 49100-000, Sergipe, Brazil; raphaelctba20@hotmail.com; 5Programa de Pós-Graduação em Nutrição Atividade Física e Plasticidade Fenotípica, Centro Acadêmico de Vitória, Vitória de Santo Antão 55608-680, Pernambuco, Brazil; claudia.lagranha@ufpe.br; 6Exercise and Immunometabolism Research Group, Postgraduate Program in Movement Sciences, Department of Physical Education, Universidade Estadual Paulista (UNESP), Presidente Prudente 19060-900, São Paulo, Brazil; fabio.lira@unesp.br; 7Faculty of Sport Science and Physical Education, University of Coimbra, 3000-456 Coimbra, Portugal; 8Núcleo de Ciências da Vida—NCV, Centro Acadêmico do Agreste—CAA, Caruaru 50670-901, Pernambuco, Brazil

**Keywords:** enriched environment, cancer, tumor growth, angiogenesis, pro-oncogenic factor

## Abstract

This systematic review aims to evaluate the influence of environmental enrichment (EE) on oncological factors in experimental studies involving various types of cancer models. A comprehensive search was conducted in three databases: PubMed (161 articles), Embase (335 articles), and Scopus (274 articles). Eligibility criteria were applied based on the PICOS strategy to minimize bias. Two independent researchers performed the searches, with a third participant resolving any discrepancies. The selected articles were analyzed, and data regarding sample characteristics and EE protocols were extracted. The outcomes focused solely on cancer and tumor-related parameters, including cancer type, description of the cancer model, angiogenesis, tumor occurrence, volume, weight, mice with tumors, and tumor inhibition rate. A total of 770 articles were identified across the three databases, with 12 studies meeting the inclusion criteria for this systematic review. The findings demonstrated that different EE protocols were effective in significantly reducing various aspects of tumor growth and development, such as angiogenesis, volume, weight, and the number of mice with tumors. Furthermore, EE enhanced the rate of tumor inhibition in mouse cancer models. This systematic review qualitatively demonstrates the impacts of EE protocols on multiple parameters associated with tumor growth and development, including angiogenesis, occurrence, volume, weight, and tumor incidence. Moreover, EE demonstrated the potential to increase the rate of tumor inhibition. These findings underscore the importance of EE as a valuable tool in the management of cancer.

## 1. Introduction

Environmental enrichment (EE) is an enhanced mental stimulation method that promotes stimuli, developing memory-demanding tasks due to the socio-environmental context where rodents can interact actively with their complex surroundings [1,2]. In this context, EE has been investigated in animal studies and considers several disorders, such as Alzheimer’s disease, Parkinson’s disease, stroke, and anxiety- and depression-like behaviors [3,4,5]. Therefore, EE is known to delay the rate of progression and/or reduce the symptoms and the severity of these diseases. Also, evidence has shown an enhancement in immune function due to EE, mitigating inflammatory disorders—a risk factor that predisposes to the development of chronic diseases, such as cancer [6].

Cancer consists of a complex of 200 diseases. It is known that a dysfunctional immune response and inflammation have a considerable impact on cancer development and its progression [7]. Moreover, studies have shown that adjuvant cancer treatment induces a permanent systemic inflammatory state, associated with high levels of inflammatory cytokines, and weaknesses in physical function [8]. Also, cancer treatment is often associated with psychological distress, chronic pain, cachexia, fatigue, and long-term impaired quality of life, all of which are related to a worse general picture of the disease and, consequently, a worse prognosis [9].

Recent studies have demonstrated the effect of EE protocols on different types of cancer. It has been shown that EE could decrease tumor growth in melanoma, colon, and intestinal cancers [10,11]. Recently, Queen et al. (2021) observed that EE mitigated Lewis lung carcinoma growth and promoted alterations in markers of proliferation and angiogenesis in lung carcinoma [12]. However, the impact of different EE protocols on cancer studies is still not completely clear. In this regard, this systematic review aims to evaluate the influence of EE on tumor factors in experimental models of cancer. We hypothesize that different EE protocols will be able to promote positive changes in parameters related to tumor growth and development.

## 2. Methods

The review followed the Preferred Reporting Items for Systematic Reviews and Meta-Analysis (PRISMA) guidelines. This systematic review was not registered.

### 2.1. Study Selection and Eligibility Criteria

Eligibility criteria were previously selected to minimize the risk of bias. The inclusion and exclusion criteria followed the PICOS (Population/Intervention/Control/Outcomes/Study) strategy (Table 1). There were no restrictions on language or publication date. Articles that did not meet the following eligibility criteria were excluded: (a) studies that used only mice and rats from different species; (b) animals subjected to intervention with different protocols of environmental enrichment; (c) those without a control group; and (d) studies that did not perform a model of cancer and did not evaluate tumor parameters. Exclusions also applied to review publications, letters, duplicates, and the presence of data used in different studies. 

### 2.2. Information Sources and Search Strategy

The search strategy was carried out during the period from December to January 2023. The databases used were PubMed (Medline), Scopus, and Embase. The search strategies used for PubMed (Medline), Embase, and Scopus were: ((“Environmental Enrichment”) OR (“Enriched Environment”)) AND ((((((Tumor) OR (Tumors)) OR (Cancer)) OR (Cancers)) OR (Neoplasm)) OR (Malignant Neoplasms)). Filters were also used in the databases: [Species, Animals, and Type of publication].

### 2.3. Selection and Data Collection Process

The screening of studies was performed by reading the title, abstract, and full text. The selection of studies was performed by two independent researchers (MSSF and GCJS), with discrepancies resolved by a third rater. Data were extracted by two independent researchers, and any discrepancies were resolved by a third rater (TOF), Figure 1.

### 2.4. Items

Data were extracted about the study (Author and year), Animal characteristics (Species, sex, age), information on the number of animals per cage; Environmental Enrichment Protocol, and Housing dimensions (Length, Width, and Depth or Height, in centimeters or meters). In addition, the exposure time of the environmental enrichment in weeks was collected. In the absence of information, data were not considered. Data were collected on cancer and tumor outcomes, including type of cancer; cancer model description; angiogenesis; tumor occurrence; tumor volume (%, cubic millimeters); tumor weight (milligrams, grams); mice with tumor and inhibition (%); and tumor size (square millimeters).

### 2.5. Methodological Quality Assessment

The SYRCLE’s strategy was used to assess the methodological quality of the animal studies. The tool consisted of ten questions that evaluated methodological criteria: Q1. Was the allocation sequence adequately generated and applied? Q2. Were the groups similar at baseline, or were they adjusted for confounders in the analysis? Q3. Was the allocation to the different groups adequately concealed? Q4. Were the animals randomly housed during the experiment? Q5. Were the caregivers and/or investigators blinded by the knowledge of which intervention each animal received during the experiment? Q6. Were animals selected at random for outcome assessment? Q7. Was the outcome assessor-blinded? Q8. Were incomplete outcome data adequately addressed? Q9. Are reports of the study free of selective outcome reporting? Q10. Was the study free of other problems that could result in a high risk of bias? Questions were answered with options of ‘Yes’, ‘No’, or ‘Not clear’. When the answer was ‘yes’, a score was given; when the answer was ‘no’ or ‘not clear’, no score was given. The overall scores for each article were calculated as a score of 0–10 points, with the quality of each study being classified as high (8–10), moderate (5–7), or low (<5). The two reviewers independently reviewed all the included studies. Discrepancies between evaluators were resolved by consensus. The quality outcomes are described in Table 2.

## 3. Results

### 3.1. Search Results

In an initial search, 770 articles were found, of which 346 duplicates were excluded. This was performed with the help of the EndNote software X20 version. Then, 424 articles were screened and subjected to the eligibility criteria, leading to the exclusion of 409 articles based on title and abstract reading. Fifteen studies remained for full-text reading, and of these, three studies were excluded. Finally, 12 studies were included in this systematic review (Figure 1).

### 3.2. Methodological Quality Assessment

The quality analysis of the studies is shown in Table 2. All studies showed adequate and randomized allocation with randomly selected animals. Furthermore, incomplete results were handled appropriately, free from selective results and bias. Because these studies involve intervention with a cancer model, it is not possible to consider the investigation and analysis of the results blindly. In general, all studies presented good quality criteria.

### 3.3. Study Characteristics

The included studies were published between 2010 and 2021. Regarding the characteristics of the animals, we observed that all included studies used mice of different species. Ten studies were performed exclusively with C57BL6 mice [10,12,13,14,15,16,17,20,21,22]. One study used C57BL6 and BALB/c mice [18], and another used B6C3F1 [19]. Of the included studies, eight used male mice, with ages ranging from 2 to 56 weeks. All included studies used physical and social enrichment protocols, according to the classification by Hosey and collaborators [23]. Within the characteristics of the different protocols, we observed that the number of animals per cage ranged from 4 to 25. In the objects used to establish EE, great diversity was observed (Huts; Igloos; Running Wheels; and Wood Toys, among others). The dimensions of the cages in terms of Length, Width, and Depth or Height are expressed in centimeters or meters in these variables. Finally, the duration in weeks of exposure to the EE ranged from 3 to 16 weeks (Table 3).

### 3.4. Types and Cancer Models

Different types of cancer were used within the included studies: breast cancer [14,16], lung cancer [12,20], pancreatic cancer model [17,22], melanoma [10,21], glioma [15], liver cancer [18], and ovarian cancer [19]. In the cancer models used in the included studies, we identified the mimicry of cancer models by injection/inoculation of cancer cells subcutaneously in nine included studies.

### 3.5. Tumor Volume, Weight, Size, and Angiogenesis after Environmental Enrichment

Eight included studies evaluated tumor volume, which is usually measured using the equation (Volume = length × width^2^ × π/6) and expressed in mm^3^ and percentage [10,12,13,14,15,16,17,18]. The results showed that EE was able to significantly reduce tumor volume in mice with colon, melanoma, breast, glioma, pancreatic, liver, and lung cancer. Additionally, eight included studies evaluated tumor weight, expressed in milligrams, grams, and percentages. [10,12,13,16,17,18,20,22]. Seven studies demonstrated a decrease in tumor weight after EE in mice with melanoma, breast, glioma, pancreatic, liver, and lung cancer [10,12,16,17,18,20,22], while only one study did not show significant differences [13]. One study only analyzed tumor size [21] and did not observe significant differences after EE. One study only analyzed angiogenesis, showing a significant reduction after EE [13] (Table 4). Therefore, it is observed that EE was able to act positively on the oncological variables analyzed in the included studies (Figure 2).

### 3.6. Tumor Number, Occurrence, Inhibition, and Mice with Tumor after Different Protocols of Environmental Enrichment

One study analyzed tumor numbers and did not identify significant differences after environmental enrichment [13]. Four studies evaluated the occurrence of tumors in mice, and a significant reduction was observed after EE in melanoma, breast, liver, and lung cancer models [10,14,18,20]. On the other hand, two studies only evaluated the percentage inhibition of tumors [17,22]. In this regard, an increase in the percentage of tumor inhibition was observed after the intervention with EE. Three included studies analyzed the percentage of mice affected by tumors [10,15]. Two studies did not observe significant differences in the prevalence of tumors [13,15]. However, one study only observed a reduction in the number of tumors in mice after environmental enrichment in an ovarian cancer model [19]. Thus, it is evident that exposure to EE was able to significantly reduce the number of tumors and their occurrence, as well as increase tumor inhibition in mice (Figure 2).

## 4. Discussion

The findings of this systematic review qualitatively demonstrate that the implementation of EE protocols has a significant impact on multiple parameters associated with tumor growth and development. Notably, EE was found to reduce tumor volume, weight, angiogenesis, tumor cell number, occurrence, and the presence of tumors in mice. These observations have important implications for understanding the possible antitumor potential of EE. Tumor volume and weight are widely recognized as reliable indicators of tumor progression and treatment efficacy in different tumor types, including colon, breast, glioma, pancreatic, liver, and lung cancers [24]. The qualitative synthesis carried out in this systematic review suggests that EE may exert its antitumor effects through hormonal mechanisms, including a decrease in cortisol levels and the modulation of sympathetic activity. It is known that the cancer initiation process is intrinsically linked to different genetic, environmental, and psychophysiological mechanisms, including chronic stress. This is responsible for the activation of inflammatory signaling, as well as heightened activity of the neuroendocrine axis, resulting in excessive production of cortisol, which can act as a trigger for tumor growth and development [25].

Studies by Kamiya et al. (2021) [24] have highlighted the significant role of sympathetic nerves in cancer progression, particularly under conditions of stress. Evidence has shown that sympathetic nerves must innervate the tumor microenvironment, promoting cancer cell alterations and angiogenesis, while also altering immune cells’ response to the cancer environment. In this regard, experimental studies have observed that chemical and surgical sympathectomy might suppress carcinogenesis from tumor initiation and its progression [26]. Moreover, Magnon et al. (2013) found that genetic deletion of stromal β2- and β3-adrenergic receptors can prevent prostate cancer development in early phases [26]. This underscores the importance of stress management and the regulation of sympathetic pathways as potential strategies to exert antitumor effects [24]. Furthermore, the ability of EE to influence tumor volume makes it a potential tool for monitoring the effectiveness of radiotherapy, as suggested by Dubben et al. (1998) [27]. No significant differences were observed in tumor size after EE.

Angiogenesis, the formation of new blood vessels, is critical for tumor establishment and maintenance. Vascular endothelial growth factor (VEGF) has emerged as a key mediator of tumor angiogenesis, promoting tumor growth, invasion, and metastasis [28,29]. Targeting VEGF and other biological agents involved in angiogenesis has been explored as a therapeutic strategy to limit oxygen and nutrient supply to tumors [30]. In this regard, EE has been shown to reduce tumor angiogenesis, suggesting its potential as a non-pharmacological approach to inhibit this process [13]. The depletion of vessels, vessel normalization, and immune activation are potential pathways through which EE may exert its anti-angiogenic effects [31]. Queen et al. (2021) noticed that EE promoted the inhibition of mRNA expression in a variety of markers of angiogenesis, such as *COX2* and *VEGF* genes [12]. Furthermore, EE has been found to regulate natural killer cells, demonstrating its immunoprotective role against cancer [32]. Encouraging the adoption of lifestyle factors such as EE, physical activity, and nutritional adjustments may further modulate tumor angiogenesis [33,34]. Besides the relevance that tumors exhibited alterations in angiogenesis due to EE, the mechanism by which EE promotes the inhibition of angiogenesis is unknown.

This systematic review also examined the effects of EE on tumor cell number, occurrence, inhibition, and the presence of tumors in mice. Consistent with the present findings, Di Castro et al. (2021) reported a reduction in glioma tumor cell proliferation in mice exposed to EE [35]. This effect may be mediated by a decrease in GABAergic activity in the peritumoral area [36]. The neurotransmitter GABA, derived from B cells, has been shown to promote the differentiation of monocytes into anti-inflammatory macrophages and inhibit the action of CD8+ T cells [36]. Therefore, the inactivation of GABAergic activity appears to enhance antitumor responses. Additionally, increased brain-derived neurotrophic factor (BDNF) production stimulated by EE has been linked to reduced tumor growth in gliomas. BDNF expression has also been associated with the enhanced maturation of natural killer cells in various tissues [37]. The findings of this systematic review highlight the beneficial effects of EE protocols on various parameters related to tumor growth and development. EE demonstrates the potential to reduce tumor volume, weight, angiogenesis, and tumor occurrence.

### Strengths and Limitations

The present study stands out as the first systematic review to compile the impacts of EE protocols on different parameters related to tumor growth and development in experimental models of cancer. Furthermore, evidence from animal models demonstrates that non-pharmacological tools, including EE, could be used as adjuvant treatments for various types of cancer. The connection between tumor growth and development and psycho-physiological stress justifies the use of EE, which is increasingly being recognized as a low-cost, viable, reproducible, and reliable tool. It can promote voluntary stimulation of cognitive, motor, and somatosensory domains, culminating in the reduction of stress-linked signaling pathways, including the pituitary-adrenal axis, through the regulation of sympathetic nervous system activation. The synthesis of statistical data is necessary for an effective understanding of the different effects, mechanisms, and responses produced by EE in the effective regulation of tumorigenesis, with the aim of developing new therapeutic strategies. Within the limitations presented by this systematic review, we highlight the possible influence of the sex and age of the animals, as well as the low variability of the rodent species used in the included studies, which mainly used C57BL6. In future studies, we highlight the importance of making comparisons between different sexes and age groups, as well as considering a greater diversity of species, which would help us understand the impacts of EE and cancer within the diversity and individuality of each organism, leading to the observation of divergences and similarities in each regulatory mechanism of tumor development.

## 5. Conclusions

In conclusion, our findings strongly support the effectiveness of environmental enrichment (EE) protocols in mitigating various parameters associated with tumor growth and development. Through EE interventions, we observed significant reductions in angiogenesis, tumor occurrence, volume, weight, and the number of cells in mice with tumors. Notably, EE also demonstrated the potential to enhance tumor inhibition rates. These findings offer valuable insights into the therapeutic landscape, suggesting that incorporating EE protocols alongside conventional cancer treatments may yield substantial benefits. EE can potentially complement existing therapies by providing a non-pharmacological means to regulate tumor growth and progression. Moving forward, further research is warranted to unravel the underlying molecular mechanisms through which EE exerts its antitumor effects. Exploring these pathways will enable the development of more targeted and personalized EE interventions, enhancing their clinical efficacy and applicability in translational studies, thus better understanding the responses of environmental enrichment in different models and types of studies.

## Figures and Tables

**Figure 1 ijms-24-16516-f001:**
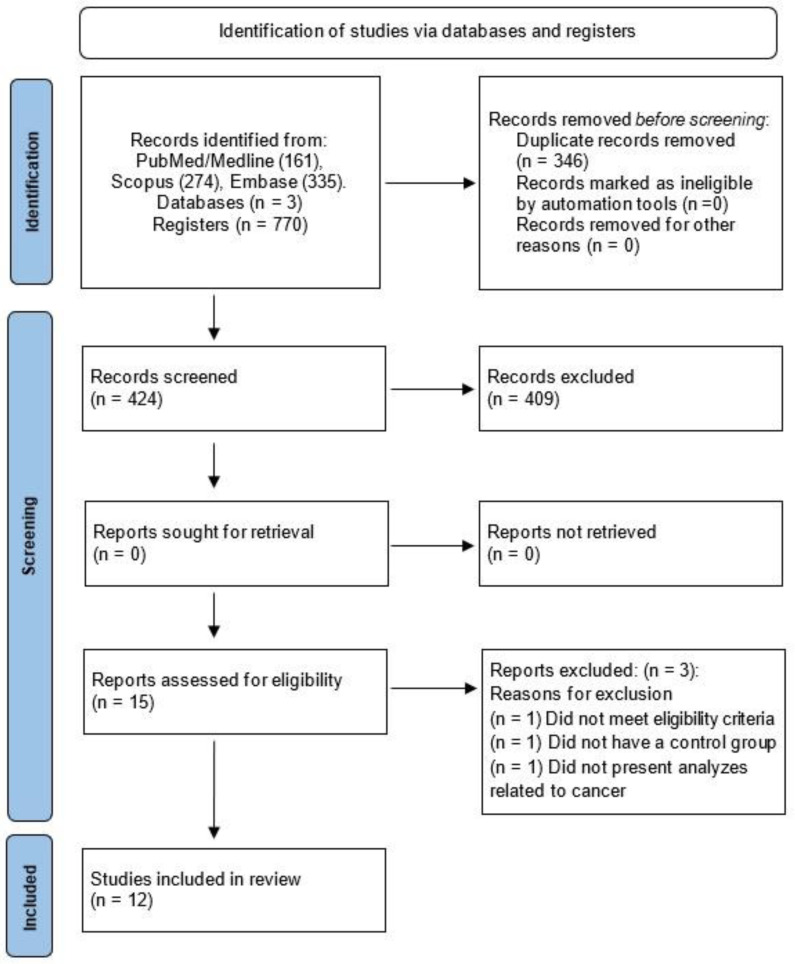
PRISMA 2020 flow diagram for new systematic reviews, which included searches of databases and registers only.

**Figure 2 ijms-24-16516-f002:**
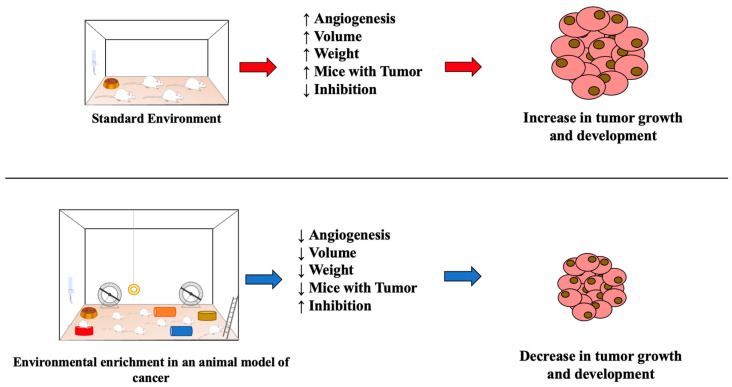
Impacts of environmental variability (standard environment—arrows in red and environmental enrichment—arrows in blue) on parameters related to tumor growth and development in experimental models of cancer. ↓ Significant decrease; ↑ Significant increase.

**Table 1 ijms-24-16516-t001:** PICOS strategy.

	Inclusion Criteria	Exclusion Criteria
Population	Rodents	Non-rodents
Intervention	Environmental enrichment	Non-environmental enrichment
Control	Non-environmental enrichment	Any other comparison group
Outcomes	Type of cancer; cancer model description; angiogenesis; tumor occurrence; tumor volume (%, cubic millimeters); tumor weight (milligrams, grams); mice with tumor and inhibition (%); tumor size (square millimeters)	No tumor parameters
Study design	Animal studies	Reviews; case reports; letters to the editor; comments, etc.

**Table 2 ijms-24-16516-t002:** Methodological quality assessment.

Author, Year	Q1	Q2	Q3	Q4	Q5	Q6	Q7	Q8	Q9	Q10
Bice et al., 2017 [13]	Y	Y	Y	Y	N	Y	N	Y	Y	Y
Cao et al., 2010 [10]	Y	Y	Y	Y	N	Y	N	Y	Y	Y
Foglesong et al., 2019 [14]	Y	Y	Y	Y	N	Y	N	Y	Y	Y
Garofalo et al., 2014 [15]	Y	U	Y	Y	N	Y	N	Y	Y	Y
Nachat-Kappes et al., 2012 [16]	Y	Y	Y	Y	N	Y	N	Y	Y	Y
Li et al., 2015 [17]	Y	Y	Y	Y	N	Y	N	Y	Y	Y
Liu et al., 2021 [18]	Y	Y	Y	Y	N	Y	N	Y	Y	Y
Queen et al., 2021 [12]	Y	Y	Y	Y	N	Y	N	Y	Y	Y
Takai et al., 2016 [19]	Y	Y	Y	Y	N	Y	N	Y	Y	Y
Watanabe et al., 2019 [20]	Y	Y	Y	Y	N	Y	N	Y	Y	Y
Westwood et al., 2013 [21]	Y	U	Y	Y	N	Y	N	Y	Y	Y
Wu et al., 2016 [22]	Y	Y	Y	Y	N	Y	N	Y	Y	Y

Legend: Q1. Was the allocation sequence adequately generated and applied? Q2. Were the groups similar at baseline, or were they adjusted for confounders in the analysis? Q3. Was the allocation to the different groups adequately concealed? Q4. Were the animals randomly housed during the experiment? Q5. Were the caregivers and/or investigators blinded by the knowledge of which intervention each animal received during the experiment? Q6. Were animals selected at random for outcome assessment? Q7. Was the outcome assessor blinded? Q8. Were incomplete outcome data adequately addressed? Q9. Are reports of the study free of selective outcome reporting? Q10. Was the study apparently free of other problems that could result in a high risk of bias? Y, Yes; N, No; U, Unclear.

**Table 3 ijms-24-16516-t003:** Sample description and characterization of environmental enrichment protocols.

Author, Year	Species, Sex, and Age	Animals per Cage	Environmental Enrichment Protocol and Housing Dimensions (Type of Enrichment, Length, Width, and Depth or Height)	Exposure Time to Environmental Enrichment
Bice et al., 2017 [13]	C57BL6 mice; female and male; 16 wks old	15–20	Physical and Social Enrichment; Huts; Mouse Igloos; Rafters; Running Wheels; and Tunnel; 15 cm × 20 cm × 29 cm	Short and long term (16 wks)
Cao et al., 2010 [10]	C57BL6 mice; male; 3 wks old	18–20	Physical and Social Enrichment; Igloos; Huts; Maze; Nesting Material; Retreats; Running Wheels; Tunnels; and Wood Toys; 1.5 m × 1.5 m × 1.0 m	3–6 wks
Foglesong et al., 2019 [14]	C57BL6 transgenic mice; female; 6 wks old	5	Physical and Social Enrichment; Igloos; Huts; Maze; Nesting Material; Retreats; Running Wheels; Tunnels; and Wood Toys; 63 cm × 49 cm × 44 cm	4 wks
Garofalo et al., 2014 [15]	C57BL6 mice; male; 3 wks−2 months old	10	Physical and Social Enrichment; Climbing Ladders; Seesaws; Running Wheel; Balls; Plastic; Wood; Cardboard Boxes; and Nesting Material; 36 cm × 54 cm × 19 cm	5 wks
Nachat-Kappes et al., 2012 [16]	C57BL6 mice; female; 3 wks old	10	Physical and Social Enrichment; Running Wheel; Tunnels; Igloos; Nesting Material; and Wooden Toys; 60 cm × 38 cm × 20 cm	16 wks
Li et al., 2015 [17]	C57BL6 mice; male; 3 wks old	12	Physical and Social Enrichment; Running Wheel; Small Huts; Tunnels; Wood Toys; and Nesting Material; 61 cm × 43 cm × 21 cm	3–5 wks
Liu et al., 2021 [18]	C57BL/6 mice; male; 2–3 wks old and BALB/c mice; male; 3 wks old	8–25	Physical and Social Enrichment; Running Wheels; Tunnels; Huts; Retreats; and Wood Toys; 40 cm × 30 cm × 20 cm	3–10 wks
Queen et al., 2021 [12]	C57BL6 mice; female; 3 and 14 months old	10	Physical and Social Enrichment; Running Wheels; Huts; Shelters; Toys; Tunnels; Maze; and Nesting Material; 120 cm × 90 cm × 76 cm	11 wks
Takai et al., 2016 [19]	B6C3F1 mice; female; 6 wks old	12–24	Physical and Social Enrichment; and Mouse Igloos; 218 mm × 320 mm × 133 mm	6 wks–100 days
Watanabe et al., 2019 [20]	C57BL/6 mice; male; 24–35 wks old	4–14	Physical and Social Enrichment; Mouse Igloos; and Fast -Track; 21.8 cm × 32 cm × 13.3 cm	10–14 wks
Westwood et al., 2013 [21]	C57BL/6 mice; male; 3 wks old	20	Physical and Social Enrichment; Exercise Wheels; Cardboard Boxes; PVC Tubes; and Plumbing T Piece; 81 cm × 57 cm × 34 cm	6 wks
Wu et al., 2016 [22]	C57BL/6 mice; male; 3 wks old	12	Physical and Social Enrichment; Exercise Wheels; Tunnels; Wood Toys; and Plastic Tubes; 61 cm × 43 cm × 21 cm	3 wks

Legend: Cm: Centimeters; m: meters; mm: millimeters; wks: weeks.

**Table 4 ijms-24-16516-t004:** Cancer types, models, and impacts of environmental enrichment on tumor outcomes in experimental models.

Author, Year	Type of Cancer	Cancer Model	Tumor Outcomes
Bice et al., 2017 [13]	Colon Tumor	Genetically induced by mutant’s phenotypes (Apc^Min^ and Apc^Min^ Tcf4^Het^)	↓ Tumor angiogenesis; Volume (mm^3^); 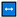 tumor number and Weight (mg)
Cao et al., 2010 [10]	Melanoma	Implanted subcutaneously in the flank (B16F10 syngeneic melanoma cell line/1 × 10^5^ per mouse).	↓ Tumor volume (% and mm^3^); tumor weight (%); 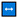 rumor occurrence (%); mice with tumor (%)
Foglesong et al., 2019 [14]	Breast Tumor	Inoculation of 50.000 breast cancer cells derived from MMTV-PyMT mice in 100 μL serum-free in the right mammary	↓ Tumor occurrence (%); volume (mm^3^)
Garofalo et al., 2014 [15]	Glioma	Were injected intracranially GL261 or CD133^+^-GL261 (7.5 × 10^4^), and U87MG (5 × 10^4^) glioma cells	↓ Tumor volume (% and mm^3^); 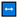 mice with tumor
Nachat-Kappes et al., 2012 [16]	Breast Tumor	Mammary cell line EO771 (56,105 cells in 100 mL) were transplanted subcutaneously into the fourth right mammary fat pad	↓ Tumor volume (mm^3^); weight (mg)
Li et al., 2015 [17]	Pancreatic	Subcutaneous tumors were prepared and implanted using Panc02 cells (6 × 10^5^ per mouse) in their right flank	↓ Tumor volume (mm^3^); weight (g); ↑ tumor inhibition (%)
Liu et al., 2021 [18]	Liver	Murine HCC cells (Hepa1-6, H22, and LPC-H12) were transplanted subcutaneously and injected into the right flanks of mice (~5 × 10^5^–1 × 10^6^) cells	↓ Tumor occurrence; volume (mm^3^); weight (g)
Queen et al., 2021 [12]	Lung	LLC cells (2.5 × 105) were implanted in mouse subcutaneous tissue with 100 μL of serum-free	↓ Tumor volume (mm^3^); weight (g)
Takai et al., 2016 [19]	Ovarian	OV3121 cells, derived from an ovarian granulosa cell tumor (5 × 10^5^ cells) were injected subcutaneously onto the back of the mice	↓ Mice with tumor (%)
Watanabe et al., 2019 [20]	Lung	3LL tumor cells (5 × 10^4^) were injected subcutaneously in the right flanks to develop solid intra-abdominal tumors. Alternatively, 3LL cells (1 × 105) were injected into the tail vein to form colonies of metastatic cells	↓ Tumor weight (g); occurrence
Westwood et al., 2013 [21]	Melanoma	Were inoculated subcutaneously with 100 μL of a single-cell suspension of 1 × 10^5^ B16F10 melanoma cells	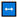 Tumor size (mm^2^)
Wu et al., 2016 [22]	Pancreatic	Panc02, Panc02-VC, or Panc02-ABCA8b cells (6 × 10^5^ per mouse) were implanted subcutaneously in the right flank	↓ Tumor weight (g), ↑ tumor inhibition (%)

Legend: g:grams; HHC: cellular hepatocellular carcinoma; LLC: Lewis lung carcinoma; mm^2^: square millimeters; mm^3^: cubic millimeters; mg: milligrams; mL: milliliters; μL: microliter; % percentage; 
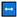
: No significant difference (*p* > 0.05). ↓ Significant decrease; ↑ Significant increase.

## Data Availability

All data from the articles included in this systematic review are available in the databases used.

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
