# Peer review of "Environmental Enrichment in Cancer as a Possible Tool to Combat Tumor Development: A Systematic Review"

_ijms, 2023, doi:10.3390/ijms242216516_

Round 1

Reviewer 1 Report

Comments and Suggestions for Authors

In this systematic review, the authors describe the effect of environmental enrichment on cancer outcomes in rodents. Appropriate methods were used to conduct the study; however, some points require attention. A major drawback of the study is that since no statistical methods, conclusions should be drawn with caution. For this reason, Lines 33-36, 39, 253-255, 258-260, 300-301, 325-327, 329, 330-332, should be modified in order for this paper to be suggested for publication.

Other specific points:

Figure 1: the legend should include an explanation for “**”

Table 4: The authors scored the studies based on the specified criteria; however, it is unclear whether this scoring was taken into consideration when presenting the data from the individual studies.

Lines 258-260: no reference to the implication of hormonal mechanisms was made in the text.

Lines 310-315: this sentence should be modified for clarity.

Lines 318-323: Another limitation is that the authors did not take into account confounding factors including sex and age of mice.

Author Response

Review 1#

In this systematic review, the authors describe the effect of environmental enrichment on cancer outcomes in rodents. Appropriate methods were used to conduct the study; however, some points require attention. A major drawback of the study is that since no statistical methods, conclusions should be drawn with caution. For this reason, Lines 33-36, 39, 253-255, 258-260, 300-301, 325-327, 329, 330-332, should be modified in order for this paper to be suggested for publication.

R-Dear reviewer, thank you for your comment. All changes in sentences related to study conclusions were made, thus demonstrating that the qualitative data of the present review indicate that environmental enrichment may be capable of modulating tumor growth and development, however, quantitative data for a meta-analysis are necessary in the future.

Other specific points:

Figure 1: the legend should include an explanation for “**”

R- Dear reviewer, thank you for your comment. In this figure, it was just a writing error, the symbol “**” was removed.

Table 4: The authors scored the studies based on the specified criteria; however, it is unclear whether this scoring was taken into consideration when presenting the data from the individual studies.

R- Dear reviewer, thank you for your comments, the questions in table 4 are part of the Joana Briggs Institute evaluation scale. All included articles are evaluated individually, generating a percentage score for each one illustrated in table 4.

Lines 258-260: no reference to the implication of hormonal mechanisms was made in the text.

R- Dear reviewer, thank you for your excellent comment, we have added the following paragraph that aims to relate the activation of the hypothalamus-pituitary-adrenal axis, cortisol production and tumor growth and development. The qualitative synthesis carried out in the present systematic review suggests that EE may exert its antitumor effects through hormonal mechanisms, including a decrease in cortisol levels and the modulation of sympathetic activity. It is known that the cancer initiation process is intrinsically linked to different genetic, environmental, and psychophysiological mechanisms, including chronic stress. This is responsible for the activation of inflammatory signaling, as well as high activity of the neuroendocrine axis that result in excessive production of cortisol, which can act as a trigger for tumor growth and development.

Lines 310-315: this sentence should be modified for clarity.

R- Dear reviewer, the text has been modified according to your request and marked in yellow. Thanks for the comment.

Lines 318-323: Another limitation is that the authors did not take into account confounding factors including sex and age of mice.

R- Dear reviewer, thanks for the comment. In our study we used age and sex in the context of the sample description, however, we recognize that these factors (age, sex) may have an influence on the context of tumor development, which is why at the end of the manuscript we added the following paragraph: "Within the limitations presented by this systematic review, we highlight the possible interference of the sex and age of the animals, as well as the low variability of the rodent species used in the included studies, which mainly used C57BL6. In future studies, we highlight the importance of making comparisons between different sexes and age groups, as well as a greater diversity of species, which would help us understand the impacts of EE and cancer within the diversity and individuality of each organism, leading to the observation of divergences and similarities in each regulatory mechanism of tumor development".  Modifications are marked in yellow.

Reviewer 2 Report

Comments and Suggestions for Authors

This article systematically reviews the impact of environmental enrichment (EE) on tumor growth and development in experimental models of cancer and sheds light on its role as a potential therapeutic condition.  It uses rigorous criteria to select eligible studies, extract data and assess quality, providing a methodological framework for future research in this area. However, the authors also acknowledge limitations such as the limited number of studies included and the lack of diversity in the animal models used, suggesting that these factors may affect the generalizability of the results. Overall, this review is adequate, but I would recommend adding a description of the following points:

(1)  Environmental enrichment (EE) efforts fall into several categories according to their methods (food-based enrichment, structural enrichment, olfactory enrichment, social enrichment, cognitive enrichment, etc. (Hosey, G et al., Zoo Animals: Behaviour, Management, and Welfare)). The specific details of the EEs in the validated articles are summarized in Table 2, under which of Hosey's categories do they fall? Also, which categories are particularly effective in cancer treatment?

(2) Please provide a checklist to demonstrate that all PRISMA checklist items have been met (http://prisma-statement.org/prismastatement/). Checklists should be attached to the paper as a supplementary table.

Author Response

Review 2#

(1)  Environmental enrichment (EE) efforts fall into several categories according to their methods (food-based enrichment, structural enrichment, olfactory enrichment, social enrichment, cognitive enrichment, etc. (Hosey, G et al., Zoo Animals: Behaviour, Management, and Welfare)). The specific details of the EEs in the validated articles are summarized in Table 2, under which of Hosey's categories do they fall? Also, which categories are particularly effective in cancer treatment?

R- Dear editor, thank you for the excellent comment. According to your request and according to Hosey's criteria, we added to the results (marked in yellow) a description of the different types of environmental enrichment used in each study. This information was also added to table 2 (marked in yellow). Furthermore, we observed that the most used were physical and social enrichment, being responsible for the greater promotion of benefits in the cancer models used in the included studies.

(2) Please provide a checklist to demonstrate that all PRISMA checklist items have been met (http://prisma-statement.org/prismastatement/). Checklists should be attached to the paper as a supplementary table.

R- Dear reviewer, thanks for the comment, we added the table with the PRISMA checklist to the end of the manuscript as requested.

Reviewer 3 Report

Comments and Suggestions for Authors

Manuscript ijms-2665644

Environmental enrichment in cancer: A possible tool to combat tumor development: A Systematic review” for International Journal of Molecular Sciences

Comments:

On what basis was the indicated cancers selection? These are tumors of different origins and properties. Can the authors conduct a better analysis by indicating the relationship between parameters such as hormone dependence of the tumor or the stage of its development? Please systematize the analysis, taking into account, in addition to the indicated general parameters, also those important from the point of view of the characteristics of specific tumors.

Author Response

Review 3#

On what basis was the indicated cancers selection? These are tumors of different origins and properties. Can the authors conduct a better analysis by indicating the relationship between parameters such as hormone dependence of the tumor or the stage of its development? Please systematize the analysis, taking into account, in addition to the indicated general parameters, also those important from the point of view of the characteristics of specific tumors.

R- Dear reviewer, thank you for the excellent comment. Our systematic review aimed to address the different effects of environmental enrichment as a tool to combat tumor development, which is why we searched the databases for a wide range of types of cancer, especially those that affect main target organs such as the liver, brain, and lung. In this way, we were able to identify the benefits that environmental enrichment can promote in central and peripheral bodies. Secondly, the present study emphasized the perspective of processes explicitly linked to tumor development, including tumor weight, tumor number/incidence, and tumor angiogenesis, in this sense it is known that hormonal factors, mainly the modulation of cortisol and its pathways, signaling. What we can observe from our review is that environmental enrichment can promote a reduction in tumorigenesis in a multifactorial way, among these factors is the decrease in cortisol secretion and reduction in sympathetic activity, increasing stress resistance, resulting in less tumor activity. Studies that observe the directionality of the relationship between hormonal factors, sympathetic activity, and environmental enrichment in cancer are still scarce. Our research group has, through the work of a research project, aimed to elucidate these possible relationships in future studies. Finally, we decided to address the different types of cancer in general, since adding an emphasis on specifics for each cancer model would deviate from the purpose of the work. Furthermore, the number of articles available for each model would be insufficient to formulate a robust explanation of the impact of environmental enrichment on each type of cancer. Additionally, what we specifically did was include in Table 1 the characteristics of each model about its cellular profile for inoculation, and cell quantity, among others that can help formulate the development of other studies. Thanks for the excellent comment and idea.

Round 2

Reviewer 1 Report

Comments and Suggestions for Authors

The resubmitted manuscript is now accepted for publication.

Reviewer 2 Report

Comments and Suggestions for Authors I agree that this article is accepted because the authors have sufficiently addressed the points raised by my peer review.

Reviewer 3 Report

Comments and Suggestions for Authors

The manuscript has been corrected and supplemented.